# Preparation and Photodegradation Properties of Carbon-Nanofiber-Based Catalysts

**DOI:** 10.3390/polym14173584

**Published:** 2022-08-30

**Authors:** Mingpan Zhang, Fuli Wang, Xinran Shi, Jing Wei, Weixia Yan, Yihang Dong, Huiqiang Hu, Kai Wei

**Affiliations:** 1National Engineering Laboratory for Modern Silk, College of Textile and Clothing Engineering, Soochow University, Suzhou 215123, China; 2Suzhou Best Color Nanotechnology Co., Ltd., Suzhou 215000, China; 3Guangzhou Inspection Testing and Certification Group Co., Ltd., Guangzhou 511447, China

**Keywords:** Fe_2_O_3_, carbon nanofibers, heterogeneous Fenton, methyl orange (MO)

## Abstract

In this study, an iron oxide/carbon nanofibers (Fe_2_O_3_/CNFs) composite was prepared by a combination of electrospinning and hydrothermal methods. The characterization of Fe_2_O_3_/CNFs was achieved via scanning electron microscopy (SEM), infrared spectroscopy (IR), X-ray diffraction (XRD) and X-ray photoelectron spectroscopy (XPS). It is shown that when the hydrothermal reaction time was 180 °C and the reaction time was 1 h, the Fe_2_O_3_ nanoparticle size was about 90 nm with uniform distribution. The photodegradation performance applied to decolorize methyl orange (MO) was investigated by forming a heterogeneous Fenton catalytic system with hydrogen peroxide. The reaction conditions for the degradation of MO were optimized with the decolorization rate up to more than 99% within 1 h, which can decompose the dyes in water effectively. The degradation process of MO by Fenton oxidation was analyzed by a UV-visible NIR spectrophotometer, and the reaction mechanism was speculated as well.

## 1. Introduction

Dyes are commonly used in modern industries, such as textile, food, paper, printing, leather, and cosmetics. They will cause serious pollution problems and bring risks to human health, such as carcinogenesis and kidney dysfunction, if discharged directly into the natural environment without treatment [1,2,3]. Therefore, how to remove organic dyes from wastewater effectively has become a hot research issue in recent years [4]. 

Currently, the treatment of printing and dyeing wastewater can be divided into adsorption technologies [5,6,7], advanced oxidation technologies, such as Fenton oxidation [8,9,10,11,12,13] and ozone oxidation [14,15], biological technologies, such as bacterial [16,17], fungal [18,19], and algal [20] treatment, membrane separation technologies, such as ultrafiltration, nanofiltration, reverse osmosis and electrodialysis [21,22], electro-Fenton, anodicoxidation [23], electrocoagulation electrochemical treatment techniques, ion exchange [24,25,26] and some multiple processes [27,28,29,30,31]. Fenton oxidation is reported to be one of the most widely studied and applied advanced oxidation technologies. It can completely degrade the refractory toxic and hazardous organic compounds into water and carbon dioxide with a short treatment cycle.

The conventional Fenton technique uses ferrous salts, but some iron-containing oxides, such as Fe_2_O_3_, can be used instead of ferrous salts. It was shown previously that composites containing Fe_2_O_3_ can effectively degrade a large number of organic compounds in wastewater [32,33,34,35,36,37,38]. There are kinds of methods that could prepare Fe_2_O_3_, mainly the co-precipitation method, solution-gel method, micro-emulsion method, solvothermal method, hydrothermal method, etc. Among them, the hydrothermal method can produce nano Fe_2_O_3_ with high purity, controllable morphology and particle size under mild synthesis conditions. Electrospinning is a simple technique for the effective production of nanofibers. The device is shown in Figure 1. Since carbon nanofibers (CNFs) have a high specific surface area, few structural defects, low density and high conductivity, they can be used as a catalytic template to prevent the agglomeration of nanoparticles which can solve the problem that nano Fe_2_O_3_ is prone to agglomeration. In this study, CNFs were obtained after the carbonization of polymethyl methacrylate/polyacrylonitrile (PAN/PMMA) nanofibers and used as template materials. In addition, Fe_2_O_3_/CNFs composites were prepared by the hydrothermal method with high purity, controlled morphological particle size, and uniform dispersibility. Then, the Fe_2_O_3_/CNFs composites were used in the Fenton reaction to degrade the MO solution.

## 2. Experimental Methods

### 2.1. Materials

Polyacrylonitrile (PAN) (Mw 150,000) was purchased from Shanghai Maclean. Polymethyl methacrylate (PMMA) (high flow type) was purchased from Shanghai Maclean. N, N-dimethylformamide (DMF) (AR) was purchased from Shanghai SiXin Bio. Ferric chloride hexahydrate (FeCl_3_·6H_2_O) (EP) was purchased from 3A chemicals. Urea (AR) was purchased from Genbio. Glycine (AR) was purchased from Genbio. Polyethylene glycol 1000 monomethyl ether was purchased from Aladdin. Ethanol (AR) was purchased from Shanghai Chemical Reagent Procurement and Supply Station. Methyl orange (MO) (AR) was purchased from Shanghai Aladdin Technology & Biochemistry. All reagents and solvents were used without further purification.

### 2.2. Preparation of Carbon Nanofiber Membrane

PAN and PMMA were added to DMF for dissolution (the mass ratio of PAN and PMMA was 4:6, and the concentration of the spinning solution was 20%), and the spinning solution was prepared by stirring at room temperature for 24 h. The spinning solution was ultrasonicated for 2 h and spun by an electrospinning machine. The spinning time was 24 h. Then, the above nanofiber film was put into a tube furnace and pre-oxidized in the air for 1 h. After the pre-oxidation, it was continued to be carbonized in nitrogen for 1 h. After the tube furnace was naturally cooled to room temperature, carbon nanofibers were obtained.

### 2.3. Preparation of Fe_2_O_3_/CNFs Composites

The Fe_2_O_3_/CNFs were prepared by the hydrothermal method. Firstly, 0.16 g FeCl_3_·6H_2_O, 0.2 g polyethylene glycol, 0.484 g urea and 0.03 g glycine were added to 25 mL deionized water. The mixed solution was stirred until it was completely dissolved and then poured into the hydrothermal reactor. Then, 20 mg carbon nanofiber membranes were added to the hydrothermal reactor to make them uniformly dispersed in the solution. The reaction was carried out at a certain temperature for a while to obtain Fe_2_O_3_/CNFs composites. 

To explore the influence of hydrothermal reaction time and temperature on iron oxide particles, the following experiments were set up:(1)The hydrothermal reaction time was 1 h, and the reaction temperatures were 130 °C, 140 °C, 150 °C, 160 °C, 170 °C and 180 °C, respectively.(2)The hydrothermal reaction temperature was 180 °C, and the reaction time was 1 h, 2 h, 3 h, 4 h, 5 h and 6 h, respectively.

Under visible light conditions, Fe_2_O_3_/CNFs were used as a catalyst for the degradation of MO.

### 2.4. Characterization

The surface structures of the composites were observed by scanning electron microscopy (SEM, s-4800) (Hitachi Manufacturing, Tokyo, Japan). The crystal structure of the composites was characterized by X-ray diffraction (XRD) (X’ Pert-Pro MRD, Panaco, The Netherlands). The functional groups of the composites were determined by infrared spectroscopy (IR) (Nikoli Instruments Manufacturing, Waltham, Massachusetts, USA). The chemical composition of the composites was examined by energy dispersive spectroscopy (EDS) (Hitachi Manufacturing, Tokyo, Japan) and by X-ray photoelectron spectroscopy (XPS) using Axis Ultra HAS equipment.

### 2.5. Degradation Experiments

A total of 100 mg/L MO solution was prepared, and the pH of the solution was 7.4. Then the degradation experiments were carried out as follows:(1)Effect of reaction temperature: Fe_2_O_3_/CNFs composite is 0.8 g/L, the concentration of hydrogen peroxide is 0.194 mol/L, and the reaction time is 0~120 min. The reaction temperatures are 50 °C, 60 °C, 70 °C and 80 °C, respectively.(2)Effect of the amount of Fe_2_O_3_/CNFs composite: The initial concentration of hydrogen peroxide is 0.194 mol/L, the reaction temperature is 80 °C, and the reaction time is 0~120 min. The amounts of the Fe_2_O_3_/CNFs composite are 0.4 g/L, 0.6 g/L, 0.8 g/L and 1.0 g/L, respectively.(3)Effect of initial concentration of hydrogen peroxide: Fe_2_O_3_/CNFs composite is 0.8 g/L, the reaction temperature is 80 °C and the reaction time is 0~240 min. The initial concentrations of hydrogen peroxide are 0.097 mol/L, 0.146 mol/L, 0.194 mol/L, and 0.243 mol/L, respectively.

### 2.6. Degradation Performance of Fe_2_O_3_/CNFs 

MO solutions with the concentration of 4 mg/L, 10 mg/L and 40 mg/L (pH ≈ 7.4) were prepared. The UV-Vis spectrums were measured by Cary 5000 UV-vis-NIR spectrophotometer between the wavelengths of 190–600 nm so as to obtain the maximum absorption wavelength.

MO solutions with the concentrations of 4 mg/L, 6 mg/L, 8 mg/L, 10 mg/L, 20 mg/L, 30 mg/L, 40 mg/L, 50 mg/L and 60 mg/L were prepared. The absorbances of MO solutions with different concentrations at the maximum absorption wavelength were measured, and the absorbance concentration standard curve was obtained. 

After the degradation experiment, the absorbance of MO was measured at the maximum absorption wavelength. Then the solution concentration was obtained according to the standard curve of MO. The decolorization rate of MO was calculated.

Determination of the decolorization rate of MO: The absorbance of MO solutions with different concentrations at the maximum absorption wavelength was measured and the absorbance–concentration standard curve was obtained. The concentration of MO before degradation was denoted as C_0_. After the degradation experiment, the absorbance was measured at the maximum absorption wavelength. Then the solution concentration was obtained according to the standard curve, denoted as C_1_. The decolorization rate was calculated as follows: (1)$$\frac{C_{0}-C_{1}}{C_{0}}\times 100\%$$

## 3. Results and Discussion

### 3.1. Characterization of Fe_2_O_3_/CNFs

Figure 2 and Figure 3 are the SEM images of carbon nanofibers and Fe_2_O_3_/CNFs prepared under different hydrothermal reaction conditions. From the figures, it can be seen that when the reaction temperature rises to 140 °C, large amounts of nanoparticles were successfully grown on carbon nanofiber. When the temperature is lower than 150 °C, the generated Fe_2_O_3_ nanoparticles are uneven, indicating that the hydrothermal reaction temperature affects the fabrication of nanoparticles. Meanwhile, the Fe_2_O_3_ particle size gradually increased with the increase in reaction time. It was known that the specific surface area increased as the number of nanoparticles increased, and the particle size became smaller. Furthermore, the catalytic efficiency improved as the specific surface area increased. Therefore, the hydrothermal reaction temperature and time were determined to be 180 °C and 1 h. Under this reaction condition, the Fe_2_O_3_ nanoparticle size is around 90 nm and grows evenly on carbon nanofibers.

Figure 4 shows the EDS element analysis chart of the Fe_2_O_3_/CNFs composites. From the figure, it can be seen that the Fe elements and O elements are uniformly distributed on the carbon fibers, indicating that the Fe_2_O_3_/CNFs composites were successfully prepared by the hydrothermal method.

Figure 5 shows the XRD patterns of Fe_2_O_3_/CNFs at different reaction temperatures and times. When the hydrothermal reaction temperature raised to 140 °C, characteristic peaks of Fe_2_O_3_ appeared at 2θ = 24.2°, 33.2°, 35.6°, 40.9°, 49.5°, 54.1°, 62.4°, and 64.0°, which are consistent with the standard card mapping card JCPDS NO.33-0664, indicating the formation of Fe_2_O_3_ particles [39].

Figure 6 shows the IR spectra of CNFs and Fe_2_O_3_/CNFs composites. In the infrared spectrum of CNFs, the characteristic peaks at 3438 cm^−1^ are caused by −OH bond stretching vibration [40]. The characteristic peaks at 1140 cm^−1^ and 1537 cm^−1^ are caused by C−O−C and C=O vibration, respectively [41]. Compared with CNFs, Fe_2_O_3_/CNFs composites showed characteristic peaks at 556 cm^−1^ and 475 cm^−1^ are caused by the vibrations of Fe-O functional groups [42].

Figure 7 shows the XPS spectra analysis of Fe_2_O_3_/CNFs composites. The satellite peak of Fe_2_O_3_ at 719.2 eV is detected in Figure 7b, which shows the trivalent Fe elemental in the Fe_2_O_3_/CNFs composites. Meanwhile, there are two main peaks at 712.5 eV and 725.4 eV corresponding to Fe 2p_3/2_ and Fe 2p_1/2_, indicating the existence of Fe^2+^ and Fe^3+^ [43]. Furthermore, the energy spectrum of C 1S is shown in Figure 7c. There are three characteristic peaks. The peak at 284.5 eV is caused by the C−C bond, while the peaks at 285.8 eV and 287.9 eV represent C−O and C=O, respectively [44]. Figure 7d shows the energy spectrum of O1 s. The peak at 531.0 eV represents the O element in Fe_2_O_3_ and the peak at 532.7 eV represents the O element in CNFs [43,44]. XPS results show that Fe−O−C chemical bonds exist between Fe_2_O_3_ and CNFs. Fe_2_O_3_ particles grew in situ on the CNFs carrier, and the results were consistent with the XRD and IR analysis.

### 3.2. Heterogeneous Fenton Degradation of MO by Fe_2_O_3_/CNFs

The MO aqueous solution has the characteristics of an acid–base indicator, and its molecular structure will change with the pH value as well colors. When pH < 3.1, it is the quinone structure and the solution is red, and when pH > 4.4, it is an azo structure and the solution is yellow. The structural changes can be expressed as follows:
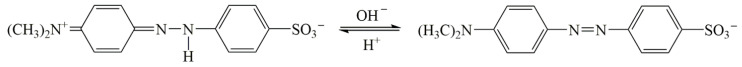


Figure 8 shows the UV spectra of MO at a different pH value. When the structure is quinone, the maximum absorption wavelength in the visible region is around 505 nm. When the structure is azo, the maximum absorption wavelength in the visible region is around 465 nm.

The decolorization effect of Fe_2_O_3_/CNFs composites on MO under different pH conditions is shown in Figure 9a. The initial pH of the solution has a great influence on the Fe_2_O_3_/CNFs composites catalyzing the degradation of MO by hydrogen peroxide. When pH < 3, the decolorization effect of MO is better, and the decolorization rate decreases gradually with the increase in pH. When pH is 6~8, the decolorization rate gradually increases. Furthermore, when pH > 8, the decolorization rate continues to decrease again. The best decolorization of methyl orange was achieved when pH = 2. It may be that the azo bond in the MO molecular structure changes to a quinone structure under the condition of pH < 3, and the destruction of N = N bond leads to the instability of the MO molecular structure, which makes the catalytic reaction easier to proceed. Secondly, due to the over acid condition, Fe_2_O_3_ is partially dissolved, more free Fe^2+^/Fe^3+^ contacts with H_2_O_2_, and more ∙OH is produced, thus improving the decolorization rate of MO [45].

Figure 9b shows the effect of reaction temperature on the catalytic effect of MO. The decolorization rate of MO was improved by increasing the reaction temperature. When the temperature increased from 50 °C to 80 °C, the decolorization rate of MO increased from 66.39% to 99.87%. Moreover, the increase in reaction temperature can effectively shorten the degradation time.

The heterogeneous Fenton reaction occurs on the surface of the catalyst, and the amount of catalyst is an important factor, affecting the decolorization effect. The catalytic effect of Fe_2_O_3_/CNFs amount on MO is shown in Figure 9c. When the amount of catalyst increased from 0.4 g/L to 0.8 g/L, the decolorization rates of MO increased from 78.5% to 97.31%. Increasing the dosage of Fe_2_O_3_/CNFs has a facilitating effect on the decolorization of MO. This is because increasing the amount of catalyst can increase the number of active sites on the surface of the catalyst and accelerate the decomposition rate of H_2_O_2_ to produce ∙OH. When the amount of catalyst was increased from 0.8 g/L to 1 g/L, the decolorization rate of MO increased slightly because the excessive catalyst will reduce the H_2_O_2_ adsorption per unit area [46,47].

Figure 9d shows the degradation efficiency of MO at different H_2_O_2_ concentrations. It can be seen from the figure that within the first 2 h when the concentration of H_2_O_2_ increases from 0.097 mol/L to 0.194 mol/L, the degradation rate of MO increases significantly. The degradation rate of MO is related to the amount of ·OH. The higher the hydrogen peroxide concentration, the more that ·OH is produced, and the degradation rate of MO increases. Continuing to increase the concentration of H_2_O_2_, the degradation rate of MO is no longer increased significantly.

It can be observed that the decolorization rate can be up to more than 99% for 100 mg/L MO solution by the Fe_2_O_3_/CNFs catalyst. This result was also compared with previous studies reported for the catalytic degradation of MO shown in Table 1.

### 3.3. Degradation Mechanism Analysis

The UV-Vis spectra of MO solution before and after degradation are shown in Figure 10, from which it can be seen that the absorption peak of MO disappeared, and no other new peaks generated. There are two possible reasons for this phenomenon: (1) the intermediate products of catalytic degradation of MO have no absorption in the range of 190–600 nm; (2) MO is directly degraded to CO_2_ and H_2_O without any intermediate products generated.

Fe_2_O_3_/CNFs composites degrade the MO solution under acidic conditions (pH = 2) when MO is a quinone structure and the hydroxyl radical HO∙ plays a major role in the degradation process. Combined with the Fenton reaction system, it is known that MO is not directly degraded to CO_2_ and H_2_O, but certain intermediate products are produced that have no absorption between 190 and 600 nm. It is speculated that there are four possible degradation pathways of MO which are shown in Figure 11. The intermediates produced by path (1) and path (2) will continue to be oxidized to other products in the subsequent reactions. Pathway (3) and pathway (4) are the main mechanisms of this degradation. The quinone structure of MO is decolorized by HO∙ oxidation, which leads to the destruction of the -N=N- group and C-N bond, thus decolorizing the MO solution. Finally, some of the intermediates are completely oxidized to CO_2_ and H_2_O as the reaction time increases.

## 4. Conclusions

In this study, Fe_2_O_3_/CNFs composites were prepared by a combination of hydrothermal and electrostatic spinning techniques. The effects of hydrothermal reaction temperature and time on the preparation of the composites were investigated. The Fe_2_O_3_/CNFs composites were successfully prepared by a hydrothermal method as proved by SEM, XRD, IR, EDS, and XPS analysis. Meanwhile, the existence of chemical interactions between Fe_2_O_3_ nanoparticles and carbon nanofibers was confirmed. When the hydrothermal reaction time was 180 °C and the reaction time was 1 h, the nanoparticle size was about 90 nm with uniform distribution. The degradation efficiency of Fe_2_O_3_/CNFs on MO was investigated under the Fenton reaction. Under the optimal reaction conditions, the decolorization rate of MO could reach more than 99% within 60 min reaction. In addition, the degradation mechanism and pathway of the reaction were also speculated.

The prepared Fe_2_O_3_/CNFs composite as a heterogeneous catalyst can be separated from water easily. Furthermore, high surface area carbon nanofibers are used as the carrier of the catalysts, which can increase the degradation property. It is known that the printing and dyeing wastewater of the textile industry contains not only dyes, but also large amounts of surfactants; therefore, the degradation performance of multiple organic pollutants will be investigated in the future work.

## Figures and Tables

**Figure 1 polymers-14-03584-f001:**
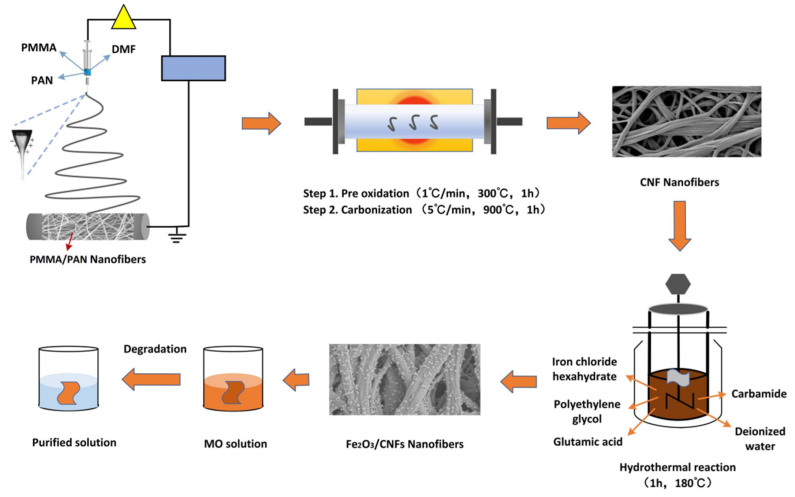
Schematic illustration of fabrication of Fe_2_O_3_/CNFs composite nanocatalysts and their degradation of MO.

**Figure 2 polymers-14-03584-f002:**
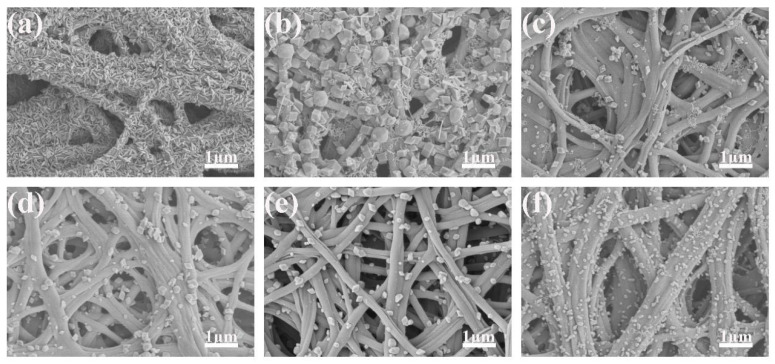
SEM images of Fe_2_O_3_/CNFs (effect of temperature): (**a**) 130 °C; (**b**) 140 °C; (**c**) 150 °C; (**d**) 160 °C; (**e**) 170 °C; (**f**) 180 °C.

**Figure 3 polymers-14-03584-f003:**
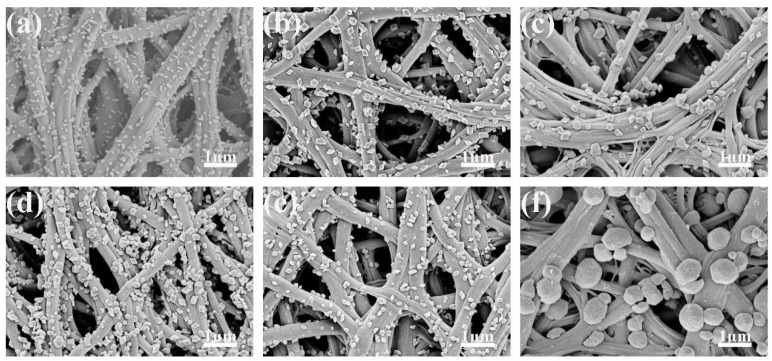
SEM images of Fe_2_O_3_/CNFs (effect of time): (**a**) 1 h; (**b**) 2 h; (**c**) 3 h; (**d**) 4 h; (**e**) 5 h; (**f**) 6 h.

**Figure 4 polymers-14-03584-f004:**
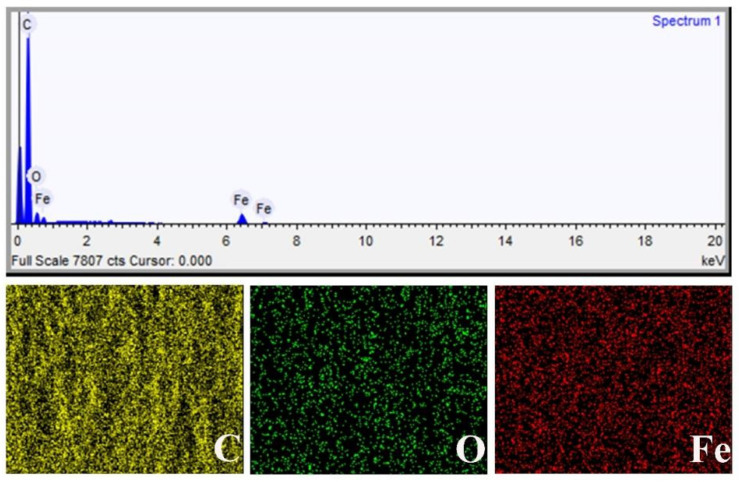
EDS image of Fe_2_O_3_/CNFs.

**Figure 5 polymers-14-03584-f005:**
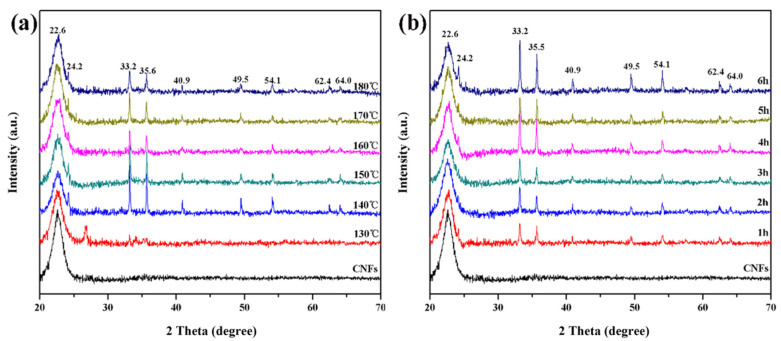
XRD patterns of Fe_2_O_3_/CNFs: (**a**) effect of reaction temperature; (**b**) effect of reaction time.

**Figure 6 polymers-14-03584-f006:**
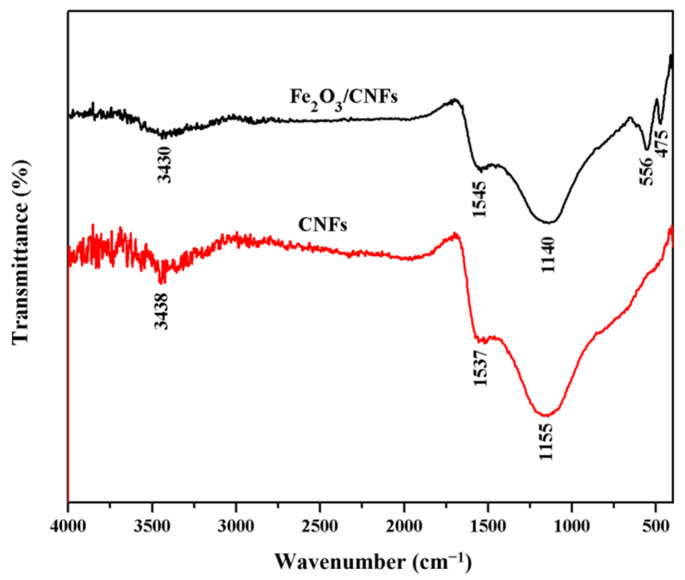
IR image of CNFs and Fe_2_O_3_/CNFs.

**Figure 7 polymers-14-03584-f007:**
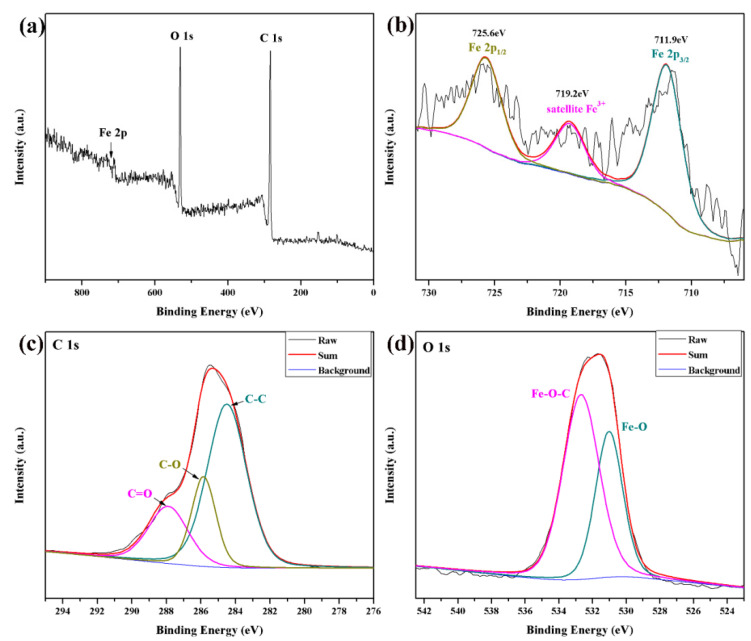
XPS spectra of (**a**) Fe_2_O_3_/CNFs; (**b**) Fe 2p; (**c**) C 1 s; (**d**) O 1 s.

**Figure 8 polymers-14-03584-f008:**
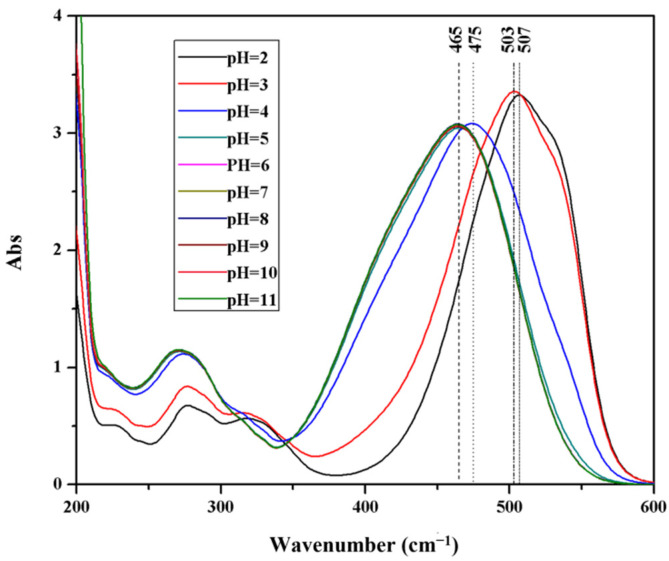
UV-vis spectra of MO at different pH value.

**Figure 9 polymers-14-03584-f009:**
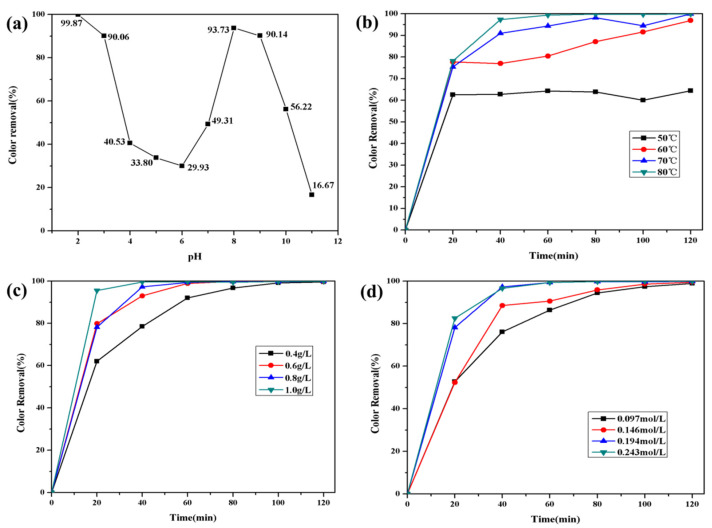
Decolorization effect of MO (**a**) with pH: ([MO]^0^ = 100 mg/L, [Fe_2_O_3_/CNFs]^0^ = 0.6 g/L, [H_2_O_2_]^0^ = 0.155 mol/L, T = 80 °C, t = 2 h); (**b**) with temperature: ([MO]^0^ = 100 mg/L, [Fe_2_O_3_/CNFs]^0^ = 0.8 g/L, [H_2_O_2_]^0^= 0.194 mol/L, pH = 2); (**c**) with Fe_2_O_3_/CNFs content: ([MO]^0^ = 100 mg/L, [H_2_O_2_]^0^ = 0.194 mol/L, pH = 2, T = 80 °C); (**d**) with H_2_O_2_ content: ([MO]^0^ = 100 mg/L, [Fe_2_O_3_/CNFs]^0^ = 0.8 g/L, pH = 2, T = 80 °C).

**Figure 10 polymers-14-03584-f010:**
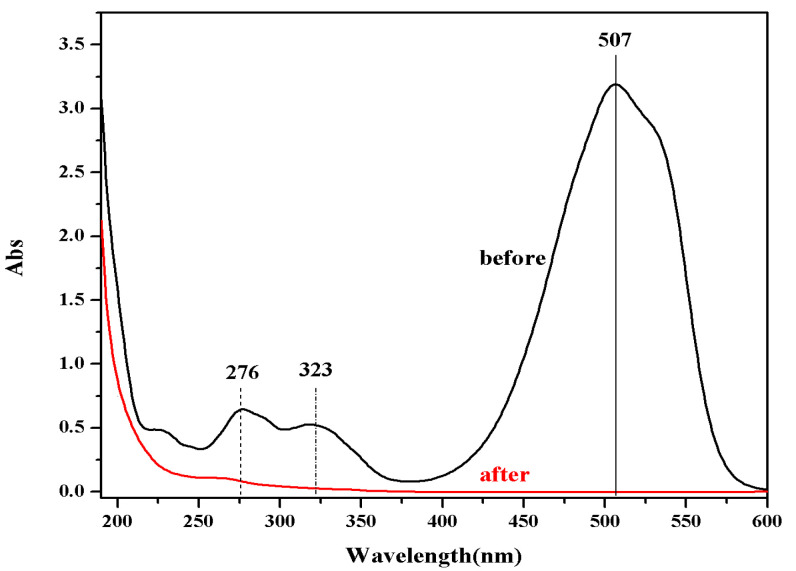
UV-vis spectra of MO before and after degradation.

**Figure 11 polymers-14-03584-f011:**
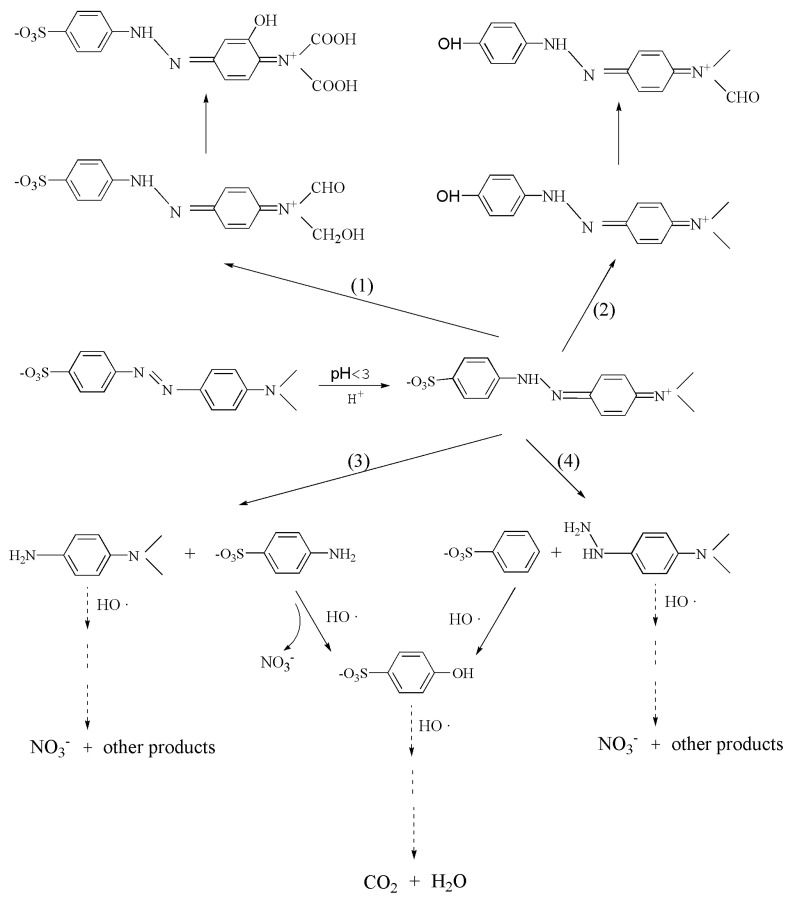
Speculation of possible degradation pathways of MO by Fe_2_O_3_/CNFs.

**Table 1 polymers-14-03584-t001:** Comparison with previous studies reported for catalytic degradation of MO.

Catalysts	Initial Concentration	Decolorization Rate	Reaction Time	References
Fe_2_O_3_/CNFs	100 mg/L	More than 99%	60 min	This work
Ti_3_C_2_-TiO_2_	40 mg/L	99%	40 min	[48]
10% Co-ZnO	100 mg/L	100%	120 min	[49]
Ag-PMOS	20 mg/L	81% & 48%	60 min	[50]
ZnO-PMOS		47% & 57%		
Ni@FP	15 mg/L	93.40%	5 min	[51]
TiO_2_/ZSM-5	20 mg/L	99.55%	180 min	[52]
PANI(1.5 mol)/ZnO	-	98.3%	180 min	[53]

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
