# Peer review of "Preparation and Photodegradation Properties of Carbon-Nanofiber-Based Catalysts"

_polymers, 2022, doi:10.3390/polym14173584_

Round 1

Reviewer 1 Report

The article is dealing with the preparation of Fe2O3/CNF materials applied in photocatalytic degradation of methyl orange (MO).

the first part of the work is focused on the preparation of the composite materials themselves and on the effect of selected experimental conditions (temperature and time of hydrotermal reaction) on the structural and textural properties of the prepared materials. I have nothing major to complain about this part of the work. The results are clear, illustrated and well described. Perhaps it would be appropriate to add to the Figures 4 and 7 which one of the prepared materials is presented on it.

However, the weakness of the publication is the second part, chapter 3.2. My comments and questions:

1.       The schema on line 180 contains H- instead of H+

2.       Nowhere is it described how the "decolorization" effect is expressed. Is it a drop in absorbance at a certain wavelength? Or a decrease in the total area of the spectrum? At what interval?

3.       Line 194: „The reason may be more ×OH can be produced under acidic conditions.“ How did the authors reach this claim? Based on the discussion with which publications? An acidic environment means a high concentration of H+ ions.

4.       The activation energy of a reaction does not depend on the temperature of the reaction. If a change in the value of the activation energy of a reaction is observed, it must be caused by a change in the mechanism of the given reaction.

MO can occur in two molecular structures: the quinone and the azo structure. This is certainly the cause of the very complex and condition-dependent mechanism of MO decomposition.

Author Response

Comments and Suggestions for Authors
(1)

The article is dealing with the preparation of Fe2O3/CNF materials applied in photocatalytic degradation of methyl orange (MO).

the first part of the work is focused on the preparation of the composite materials themselves and on the effect of selected experimental conditions (temperature and time of hydrotermal reaction) on the structural and textural properties of the prepared materials. I have nothing major to complain about this part of the work. The results are clear, illustrated and well described. Perhaps it would be appropriate to add to the Figures 4 and 7 which one of the prepared materials is presented on it.

However, the weakness of the publication is the second part, chapter 3.2. My comments and questions:

  1. The schema on line 180 contains H- instead of H+

Answer: The schema has been modified.

  1. Nowhere is it described how the "decolorization" effect is expressed. Is it a drop in absorbance at a certain wavelength? Or a decrease in the total area of the spectrum? At what interval?

Answer: Determination of the decolorization rate of MO: The absorbance of MO solutions with different concentrations at the maximum absorption wavelength were measured and the absorbance-concentration standard curve was obtained. The concentration of MO before degradation was denoted as C0. After degradation experiment the absorbance was measured at the maximum absorption wavelength. Then the solution concentration was obtained according to the standard curve, denoted as C1. The decolorization rate was calculated as following:

  1. Line 194: „The reason may be more ×OH can be produced under acidic conditions.“ How did the authors reach this claim? Based on the discussion with which publications? An acidic environment means a high concentration of H+ ions.

Answer: “It may be that the azo bond in the MO molecular structure changes to quinone structure under the condition of pH < 3, and the destruction of N = N bond leads to the instability of the MO molecular structure which makes the catalytic reaction easier to proceed. Secondly, due to the over acid condition, Fe2O3 is partially dissolved, more free Fe2+/Fe3+ contacts with H2O2, and more ∙OH is produced, thus improving the decolorization rate of MO.”

  1. The activation energy of a reaction does not depend on the temperature of the reaction. If a change in the value of the activation energy of a reaction is observed, it must be caused by a change in the mechanism of the given reaction.

MO can occur in two molecular structures: the quinone and the azo structure. This is certainly the cause of the very complex and condition-dependent mechanism of MO decomposition.

Answer: Thank you for your suggestion. The activation energy of a reaction does not depend on the temperature of the reaction and this part has been modified.

Reviewer 2 Report

The study represents the photodegradation of methyl orange dyes using a carbon nano fiber-based catalyst. It can be accepted after the following changes are carried out. 

1. The abstract should be more details with numerical findings of result. 

2. There is a lack of literature regarding dye degradations in the introduction sections. It must be improved. It is suggested to use recent publications. 

Ex: https://doi.org/10.1007/s10876-022-02280-z

https://doi.org/10.1080/03067319.2022.2079083

3. Whenever you write the abbreviation in first time, use the full meaning of it. Such as SEM, IR, XRD and XPS. 

4. The hydrothermal method has not been described well. You should include the details procedure of hydrothermal synthesis and how to control the growth of particles inside the reactor. 

5. How did you optimize the synthesis conditions? It is important to discuss the achieving of optimized conditions. 

6. When using different testing procedures, please use the appropriate references. 

7. In your result and discussion, there was no references which support your statement. Therefore, please check the every analysis and add references properly. You can see this paper for your reference: https://doi.org/10.1007/s10570-021-04401-9

8. Comparing your photodegradation result with previously published papers is better. Make a table and compare it. 

9. The author needs to demonstrate the fitted line in the kinetic study during the dye degradation. You can see this paper: https://doi.org/10.1016/j.jallcom.2021.162502

10. Discuss the conclusion highlighting the novelty of your work and future implications. 

Please note that the references I suggested are an example to help the author improve the revision quality. 

Author Response

Comments and Suggestions for Authors
(2)

The study represents the photodegradation of methyl orange dyes using a carbon nano fiber-based catalyst. It can be accepted after the following changes are carried out. 

  1. The abstract should be more details with numerical findings of result. 

Answer: The abstract has been modified.

  1. There is a lack of literature regarding dye degradations in the introduction sections. It must be improved. It is suggested to use recent publications. 

Ex: https://doi.org/10.1007/s10876-022-02280-z

https://doi.org/10.1080/03067319.2022.2079083

Answer: Yes, They have been modified.

  1. Whenever you write the abbreviation in first time, use the full meaning of it. Such as SEM, IR, XRD and XPS. 

Answer: Yes, They have been modified.

  1. The hydrothermal method has not been described well. You should include the details procedure of hydrothermal synthesis and how to control the growth of particles inside the reactor. 

Answer: The hydrothermal method has been described in detail. (chapter 2.3)

  1. How did you optimize the synthesis conditions? It is important to discuss the achieving of optimized conditions. 

Answer: To explore the influence of hydrothermal reaction time and temperature on iron oxide particles, the following experiments were set up:

(1) The hydrothermal reaction time was 1h, and the reaction temperatures were 130 ℃, 140 ℃, 150 ℃, 160 ℃, 170 ℃ and 180 ℃ respectively.

(2) The hydrothermal reaction temperature was 180 ℃, and the reaction time was 1 h, 2 h, 3 h, 4 h, 5 h and 6 h, respectively.

The optimized conditions were discussed in chapter 3.1.

  1. When using different testing procedures, please use the appropriate references. 

Answer: The references have been added.

  1. In your result and discussion, there was no references which support your statement. Therefore, please check the every analysis and add references properly. You can see this paper for your reference: https://doi.org/10.1007/s10570-021-04401-9

Answer: Thank you for your suggestion. The references have been added in result and discussion.

  1. Comparing your photodegradation result with previously published papers is better. Make a table and compare it. 

Answer: The Comparison of photodegradation results has been added.

“It can be observed that the decolorization rate can up to more than 99% for 100 mg/L MO solution by Fe2O3/CNFs catalyst. This result was also compared with previous studies reported for catalytic degradation of MO shown in table 1.”

Table 1. Comparison with previous studies reported for catalytic degradation of MO

Catalysts

Initial concentration

Decolorization rate

Reaction time

References

Fe2O3/CNFs

100mg/L

More than 99%

60min

This work

Ti3C2-TiO2

40mg/L

99%

40min

[1]

10% Co-ZnO

100mg/L

100%

120min

[2]

Ag-PMOS

20mg/L

81% & 48%

60min

[3]

ZnO-PMOS

47% & 57%

Ni@FP

15mg/L

93.40%

5min

[4]

TiO2/ZSM-5

20mg/L

99.55%

180min

[5]

PANI(1.5mol)/ZnO

-

98.3%

180min

[6]

  1. The author needs to demonstrate the fitted line in the kinetic study during the dye degradation. You can see this paper: https://doi.org/10.1016/j.jallcom.2021.162502

Answer: Thank you for your suggestion. However, the kinetic study was not discussed in this work.

  1. Discuss the conclusion highlighting the novelty of your work and future implications. 

Answer: “The prepared Fe2O3/CNFs composite as a heterogeneous catalysts can be separated from water easily. Furthermore, high surface area carbon nanofibers are used as the carrier of the catalysts which can increase the degradation property. It is known that printing and dyeing wastewater of the textile industry contains not only dyes but also large amounts of surfactants, therefore, the degradation performance of multiple organic pollutants will be investigated in the future work. ”

Round 2

Reviewer 2 Report

Acceptance for publication is recommended because the authors thoroughly updated the manuscript and addressed the issues raised by reviewers.